# Coexistence of Diversified Dog Socialities and Territorialities in the City of Concepción, Chile

**DOI:** 10.3390/ani10020298

**Published:** 2020-02-13

**Authors:** Hugo Capellà Miternique, Florence Gaunet

**Affiliations:** 1Geography Department, University of the Balearic Islands, 07122 Palma de Mallorca, Spain; 2Laboratoire de Psychologie Cognitive, Fédération 3C, Aix-Marseille University-CNRS, 13331 Marseille, France

**Keywords:** stray dog, pet dog, city, sociality, territoriality, adaptation, behavior, cognition, social, spatial, geography

## Abstract

**Simple Summary:**

Stray dogs are a common sight in cities all over the world, especially in Latin America, but research on their behavior is scarce. Studying their very considerable presence in Concepción (Chile) provided a unique opportunity to learn more about the extent of the sociality and territoriality patterns of the dog species. Interestingly, a wide range of socialities with humans (and with other dogs) were shown to be dependent on human activities and urban zoning signaled by passages, physical boundaries and landmarks. New forms of sociality were also evidenced, with dogs exhibiting intermediate degrees of sociality between the *pet* and *stray dog* categories. We postulate that this unique diversity of sociospatial positioning and level of adjustment (e.g., dogs using crosswalks either alone or with people) is made possible by the city’s specific human culture and range of urban areas. The dog species thus exhibits a considerable potential for social and spatial adjustment. The fact that it depends on the spatial layout and human culture of their environment explains the presence of dogs wherever humans are. Furthermore, it has implications for coping with the presence of dogs in numerous and varied human societies.

**Abstract:**

There has been scant research on the presence of stray dogs in cities. Studying their very considerable presence in Concepción (Chile) provided a unique opportunity to learn more about the different patterns of sociality and territoriality exhibited by the dog species. Via a set of case studies, we examined the behavior of urban dogs, adopting an ethnographic methodology. This yielded findings of the dogs’ cognitive, social and spatial adjustment abilities, i.e., their territorialities. Our hypothesis was validated: We found numerous types of sociability, we confirmed the presence of two previously established categories: *family dogs* (pets, guard dogs and beggars’ dogs) and *stray dogs* (dogs almost entirely unused to humans, aggressive dogs at the far end of the campus and feral dogs in the woods). We also identified three new ones: *familiar stray dogs in packs* (dogs both spatially and socially close to humans), *pet-stray dogs* (i.e., *village dogs* interacting closely with people) and *free-roaming pet dogs*. We conclude that an ongoing two-way bond between humans and animals allowed these dogs to became part of a city’s urban identity and explains the stray dogs’ plasticity in terms of adapting to the diversified urban habitat. We postulate that it was the human culture and range of urban areas in Concepción that gave rise to this unique diversity of sociospatial positioning and level of adjustment (e.g., dogs crossing crosswalks).

## 1. Introduction

There are between 700 million and 1 billion so-called *feral* or *village dogs* in the world today [1,2]. Among the many peculiarities of Latin American cities, but one that is rarely mentioned, is the omnipresence of dogs. This is so marked that these dogs have become an intrinsic and almost picturesque feature of the urban habitat, particularly in the cities of Chile (e.g., [3,4]), where dogs are commonly referred to as quiltros.

### 1.1. Background: Research on Forms of Canine Territoriality

At first glance, the presence of dogs in Chilean cities is reminiscent of the presence of dogs in the city of Istanbul (human population around 15 million, stray dog population around 150,000, with an additional 30,000 in shelters [5]), monkeys in some Asian cities (e.g., [6,7]) and indeed other carnivores in various anthropogenic environments (e.g., [8,9]). Chilean dogs are, however, less numerous than free-ranging dogs in countries such as India [10].

The present study explored the various forms of behavioral plasticity that allow dogs to live in urban and social environments [11,12], based on the territoriality concept used in geography (Appendix A for the territoriality concept in geography).

To understand dogs’ spatial adjustment, we drew on the work by Wolch and Emel [13] in several cities in the United States, and more particularly on Beck’s investigations in the early 1970s of stray dogs in Baltimore [14,15,16,17,18]. We also drew on similar studies conducted in Europe, including Valencia in Spain [19] and Lyon in France [20]. Most of the recent studies of free-ranging dogs were conducted in India, and either focused on their cognition (e.g., [21]) or adopted a quantitative approach to their free behaviors (e.g., [10]). Whereas ethological studies have a major statistical and quantitative component, the present study adopted an ethnographic and qualitative approach to examine the behavioral adjustment of dogs in an urban context. An ethological study [22] with an ethnographic component [23] had earlier been undertaken among dogs in a Russian city [24]. Such studies were taken as initial cases to understand dogs’ spatial adjustment.

### 1.2. Emergence of the Various Categories of Domestic Dogs

Importantly, the canine species has evolved over thousands of years. For instance, it now thrives on a diet rich in starch [11] that is close to the human diet, allowing closeness with humans. Observations of the living conditions of dogs and how they organize themselves with respect to humans have led to the identification of the following dog categories, see [17,25] for an overview; [26,27,28]:-*restricted dogs*, which are totally dependent upon people: all their essential needs are intentionally met by humans, who also restrict their movement;-*family dogs*, which have owners on whom they depend, although they may be free to roam. Their reproduction is supervised by humans;-*stray dogs*, which include dogs living in a human-dominated context. This is a heterogeneous group: it includes dogs that still have a social bond with humans, possibly abandoned or born in human settings, and which tend to associate food with particular households [29,30], even when the homeowners claim that the dogs do not belong to them (Type I) [31] and dogs that exhibit varying degrees of fear/tolerance toward humans (Type II). These dogs are attracted to human settings by the availability of food and shelter, regardless of whether these resources are intentionally provided by humans or just casually associated with them. They have close proximity to humans, and beg from them rather than scavenging [32]. These two types of stray dogs, with two possible levels of relationship toward humans, are sometimes referred to as *village dogs* [31,33]. They are found roaming in many of the large cities of the Mediterranean basin (Istanbul, Alexandria);-*feral dogs*, which include all dogs living in a wild and free state with no direct food or shelter intentionally supplied by humans, showing no evidence of socialization with but rather avoidance of humans [17,34,35,36] and occupying mainly natural environments.

The distinction between feral and stray/neighborhood/village dogs is a matter of degree [36,37], see the dogs in Moscow in the movie by Riolon [38] based on the work by Andrei Poyarkov. Dogs may also change categories in the course of their lives. Social adjustment, based on learning, is one of the most distinctive cognitive features of the species, as cognitive ethology regularly demonstrates [1,39,40,41,42,43,44,45,46]. For instance, it has been shown [19] that when dogs are abandoned, they organize themselves into packs, exhibiting the more sociable behaviors of the species [47]. This supports the notion that dogs can display various forms of sociality in the same space and time, with conspecifics, humans and probably other species as well [26].

The presence of free-roaming/stray dogs has attracted the attention of ethologists working in both rural (e.g., mountainous Abruzzo region of Italy [33]) and urban (e.g., Valencia [19]; India [48,49]; Saint Louis, US: [15] and other American cities: [50]) contexts. The behavior of these dogs and their distance from humans depend largely on the nature of human activities in the different urban areas, and probably also on human cultures and practices, as their density varies from city to city. These factors may affect the closeness or otherwise of intra- and inter-specific relationships, and thus the accessibility of food [51]. For instance, some dogs may have to scavenge in faraway landfill sites (Mexico City; for Istanbul, see Appendix B as a detailed example) or in nearby trash (Pemba in Tanzania; Morocco), while others are given food on the ground a few meters away (villages in Benin and in Mali, etc.) or are directly fed from a hand or on a plate (Lesotho). Relationships between domestic dogs and humans also vary according to whether they are used for hunting, as beasts of burden (e.g., sled pulling), as guard dogs or as pets. These relationships may also be understood in different ways, depending on how humans behave and the extent to which they share their living spaces with dogs. For instance, in the Western world, human relationships with dogs differ according to whether the dogs are in a rural or urban context. Fielding, Mather and Isaacs [52] commented that the limited way in which local people in the Bahamas interact with dogs may hinder the latter’s socialization with humans, and suggested that scant dog–human interactions may lead to roaming dogs who are shy of people.

The stray dogs of the Americas live in a specific human cultural context. First, the human–dog bond dates back to the early settlement of people in the Americas [3,53] (see Figure 1). Second, in Mesoamerican cultures, humans had a special relationship with animals in general and domestic ones in particular [54]; for possible detailed explanation of the reasons for the types of presence of these dogs in Concepción see Appendix C. However, this relationship became more controlling with the arrival of Europeans at the beginning of the 16th century, at least for certain types of dogs. Finally, it is also important to consider the continuing influence of rural ways on urban lifestyles, especially on the edges of Latin American cities [55]. Dogs are free, and sound the alarm (strangers or earthquakes, which are especially numerous in Chile). For instance, owned free-roaming dogs in the Valdivian temperate forests of Chile are used to alert humans about predators and protect livestock [56]. Referential communication abilities, a human-like ability, toward humans is used here (e.g., [57,58]). All these situations where dogs are free and form different intra- and inter-specific social organizations may contribute to their behavioral plasticity, providing that their resting, eating, reproduction and social interaction needs are fulfilled [59].

The aim of the present study was to characterize and understand the presence or behaviors of dogs in the city of Concepción, Chile. Bradshaw [1] suggested that dogs have the ability to continue reinventing themselves as human society changes. In the present case, we examined the extent to which dogs living in cities constantly adjust and reinvent themselves, in order to keep up with changing human spatial and social situations. In other words, in Concepción, where human society and its activities vary, dogs live in a diversity of ecological niches, some anthropized, others natural [14]. We thus expected to observe numerous well-established types of socialities in this city, as well as several new ones.

## 2. Materials and Methods

Concepción is the second-largest city in Chile (more than 1 million inhabitants), around 500 km south of Santiago de Chile (Figure 2). It is located in the central area of the country and has a mild climate (mean rainfall: 1100 mm/year; temperature ranging between 9.5 °C and 17.4 °C). The urban area of Concepción forms a polynuclear urban pattern (133 km^2^). The city has gradually spread out and filled the gaps in between in a sprawl pattern. Owing to successive mega-earthquakes, the city has actually been rebuilt eight times since 1550 (i.e., roughly twice a century).

The present study had a traditional ethnographic research design, but also adopted the post-structuralist approach developed in social sciences [60], based on actor–network theory [61]. The original aspects of the present study consisted of applying ethnomethodology in an interdisciplinary (natural sciences with ethology, and social sciences with cultural geography) [62] and multispecies (human and nonhuman) context [63,64], focusing also on individual cases [65,66]. We therefore used a neopraxiological [67] approach, based on a detailed description of the dogs in the city of Concepción and their behaviors (sleeping, begging, eating, etc.). This resulted in (1) a fieldwork study based on written notes [68], augmented with (2) ethological references about urban dogs, (3) information from local networks and (4) interviews with locals, in order to put the ethnographic objects and animals into an actor–network theory context.

We set out to describe the territorialities of the dogs of Concepción in terms of the areas they inhabited and the specific behaviors they displayed. Two areas were studied: Plaza Perú and adjacent areas (downtown area), and Yugoslavia street in Hualpén (suburban area). The fieldwork was undertaken by the first author (HCM), drawing on earlier observations made as a cultural geographer, witness and citizen. He had been living in the city for 10 years, walking his dog on a daily basis. The methodology consisted of ethnographic observations [69,70] of spatial behaviors and social interactions with other dogs or with humans, and the identification of individuals (either specific dogs or groups of dogs). To estimate the density of dogs, the first author counted all the dogs he encountered during the 2-week fieldwork study, in order to achieve a spatial snapshot. Firstly, the dogs were classified either as *pet dogs* if there was an owner, including guard dogs and one dog belonging to a beggar or as *free-roaming dogs* if there was no owner. Secondly, as the dogs differed greatly in color, size, the shape of ears and so on, they could each be visually identified by the first author. Qualitative observations were performed for 1 week in each of the two areas. Stray dogs in a pack, along with the pet dog Cachupín, were observed in the Plaza Perú area and the residential stray dog Snoopy was observed in Hualpén. The ethnographic fieldwork following specific stray dogs (the pets and nonpets cited above and others around Plaza Perú up to the campus and its borders) was organized on the basis of prior information about these dogs earlier gotten by HCM. The first author followed the dogs all day long, at a distance of 5–10 m. In the case of Cachupín, Mister Perro and Snoopy, general information was also collected earlier. Notes were either taken online during the observations or afterwards in a notebook, for the methodology see [61,62,66]. During the writing of the paper, the first author was further questioned by the second author (FG), a cognitive ethologist, to confirm or deepen certain observations. Lastly, findings were triangulated with local news and blogs and with interviews with neighbors or passersby, to double-check the data yielded by the empirical approach and thus avoid any subjective bias.

The methodological interest of this article, therefore, lay in the combination of a qualitative approach to data collection, related to the social scientific approach of the cultural geographer (first author) and the dual ethological and cognitive ethology approach of the second author.

## 3. Results

### 3.1. Plaza Perú and Adjacent Areas

The analysis of canine habits in an urban context began in Plaza Perú, which has a range of urban characteristics. First, Plaza Perú is a square, and adjoins an avenue and an array of spaces with disparate uses: one is a residential and commercial area (red rectangle in Figure 3), while another is a residential area next to a university campus in a park (below red rectangle in Figure 3). Second, Plaza Perú occupies a central location within the city [71]. Third, there is a large daily flow of pedestrians.

The first important finding was that there were a large number of dogs (103) in this 1-km^2^ area (i.e., 0.103 dogs per 100 m^2^), and they belonged to no fewer than eight different categories, as seen in Figure 3 and Table 1. They were characterized by specific walking speeds and directions of the walk, as well as by specific postures and gaze directions. This is probably the first time that so many different types of dogs have been found sharing such a small, albeit diversified, urban area. Previous studies had evidenced just one type of dog in each area (see Section 1.2. “Emergence of the various categories of domestic dogs”). This finding probably reflects the special history of humans and dogs in Chile and Concepción. Still, higher densities of dogs are found in countries such as India [10].

This finding points to a dual behavioral plasticity: that of dogs and that of humans. The dogs that were present displayed various types of sociality with their conspecifics and with humans, as well as various types of adaptation to their habitat (house or apartment, streets and woods). In other words, they exhibited different territorialities, insofar as they had learned to live in eight specific sociospatial situations. We should not forget that dogs have a unique ability to form attachments to humans and to other dogs during their first 4 months of life (e.g., [1]). Additionally, dogs resist well to hunger and do not tend to search far and wide for food. Rather, they tend to move in close to a source of food and wait for it to come to them [51]. These abilities may facilitate varying forms of sociality and degrees of closeness to humans and conspecifics (i.e., sociabilities). We suggest that these different forms of dog sociality have arisen as a result of the human cultural and historical context. Whereas the presence of dogs in the home and on a leash is driven by Western culture, the presence of free dogs in and around Concepción (for detailed descriptions, see Table 1) is driven by the virtual absence of local and national public health policies (see TV journals [72,73]). It is also the result of a cultural context in which dogs are viewed as animals that may not necessarily be tame or domesticated, but which can live freely around humans, using their food and shelter. Dogs are also often abandoned in an area (Lenga) far from the city center, along the end of the road at the beach, bordered by restaurants where local tourists come. They are not in such good health as those in Concepción, and clearly belong to the Type II category of stray dogs that remain at a greater distance from people than the dogs in Concepción, probably because their groups are larger and thus appear scarier. Additionally, in Concepción, people behave as though the dogs were autonomous social agents that had their own space and habits in the city. Citydwellers’ behavioral plasticity therefore consists of tolerating and managing all eight situations of dogs’ social and spatial lives, contrary to typical Western cities that only tolerate pet and guard dogs and very few free-roaming owned dogs. To this respect, see [74] for an example of how representations of dogs and therefore practices toward them influence the cooperative behavior of guide dogs.

Our second important finding updated the classification of dogs related to humans (see current classification in Section 1.2. “Emergence of the various categories of domestic dogs” [17,25,28]). Table 1 shows extremes of individuals in both the stray dog category, exhibiting different degrees of fear/tolerance (familiar stray dogs; Type II) and the pet dog category (free-roaming pet dogs), based on their social behaviors and the areas they occupied.

Regarding familiar stray dogs, no stress behaviors [75] were observed, and they exhibited very little fear, and thus a high degree of tolerance toward both regular pedestrians and unfamiliar persons. Strictly speaking, they belonged to the stray dogs Type II category, but lived in a city and displayed a high degree of proximity to unfamiliar persons, compared with the stray dogs described in earlier studies. For instance, the observations made by Coppinger and Coppinger [76] and Ortolani, Vernooij and Coppinger [31] in African villages, or by Majumder, Chatterjee and Bhadra [10] in India, revealed neutral experiences of humans (possibly close/submissive to familiar people) and flight instances ranging from 0 m to more than 5 m, resulting in scavenging behavior in proximity to humans. The observations by Dias et al. [77] on the university campus of San Paolo and by Bonanni and Cafazzo [78] around Rome highlighted consumption of leftover food and greater distance from humans (see also Alkan [79] and Fortuny [80] for Istanbul, and Font [19] for Valencia). It should be noted that pictures on the Internet showing dogs in Istanbul sheltering in the lobbies of buildings suggest that some dogs are closer to some people there. In any event, the stray dogs in Plaza Perú exhibited greater proximity to and sociality with humans than those belonging to the same category reported in the literature. Dogs could stay within 2 m or even a few centimeters of humans when resting or begging, especially when Cachupín, the free-roaming pet dog, was present; social facilitation may have been at play here [81].

Overall, very few occurrences of harassment of dogs by humans were reported, and even if dogs and humans seldom directly interacted, humans’ spatial tolerance meant that motorists avoided dogs, in contrast to what happens in most other cities (e.g., [82]). This behavior extended to pedestrians stepping around them (e.g., dogs sleeping in potholes or on the front steps of restaurants). The dogs were also given food by both unfamiliar and familiar persons, and not only entered university classrooms that opened directly outside, but even rested there. A few were accidentally taken away in police vans with demonstrators during recurrent protests and marches in the Plaza area that led law enforcement officers to use water cannons and police horses, a great attraction to the dogs. This degree of proximity has never been reported in other parts of the world. Additionally, although some dogs randomly crossed streets, they were often seen using crosswalks, either alone or with conspecifics (see Figure 4). This safe behavior was probably learned either by observing [81,83,84] or by following pedestrians. Recent results have revealed that the more affiliated dogs are with humans, the more they imitate their behavior [85,86,87]. The dogs’ behavior, therefore, reflected the pedestrians’ nonthreatening and neutral attitudes toward them, and probably also their common prosocial behaviors (e.g., to cross the road to obtain food). Using the crosswalks on their own therefore showed that the dogs had learned that the residents of Concepción were just as reliable as dog owners are (see Appendix D for videos showing this observation in other places in Chile). In light of the results reported by Duranton and colleagues [85,86,87], we suggest that these stray dogs behaved as though they were affiliated with the residents of Concepción. Dogs may assign reliability on humans’ behaviors are reliable until subsequent visual cues or odors tell them otherwise [88]. Further confirmation comes from the fact that they were appreciated and popular, referred to as quiltros, meaning stray/roaming dogs leading a carefree life like vagabonds. We, therefore, regarded them as representing an extreme of the stray dogs category (Type II), and described them as familiar stray dogs in a pack.

The stray dog pack that occupied Plaza Perú (Figure 5) comprised three or four dogs, the precise number varying depending on the day of the week or the time of day. Sometimes, the dogs even deserted the square altogether.

Macdonald [89] suggested that canids, and probably all carnivores, adjust their intraspecific organization and behavior according to the patterns of resource availability. These dogs were of different breeds, ages and sexes. The smallest dog was a female dog. Her appearance suggested that she was also the oldest; present in the area the longest and very much the leader, like the female in the small group of dogs observed by Fox [15]. All the dogs maintained considerable autonomy. Thus, even though they were all accustomed to sleeping in that area, during the day they were seen on their own in different sectors. They could reasonably be viewed as a pack, insofar as we could observe certain organizational behaviors that set the standards for sociability among carnivores [19,89]. For example, they shared the same resting space and slept together, there was no conflict between them and they joined forces to meet a common goal (e.g., protecting a nest or territory). They would go and feed by themselves, but then return to the nest and seemed to have affiliative bonds.

Regarding the two owned dogs, they used to roam a great deal and exhibited highly social behaviors toward both humans and dogs, compared with similar owned dogs in other places, see for instance the Brazilian Atlantic Forest [90] and coastal villages in Michoacán, Mexico [91].

The first dog with an owner seen in the area was an old basset hound (Figure 6) who roamed the neighborhoods. Interestingly, he used to walk on the actual roadway with the cars, but very close to the sidewalk. As we never deliberately followed him in the course of our observations, the precise nature of his activities remains unknown, but he was observed walking on roads in different places up to 1 km apart, as well as close to other dogs (Figure 6).

The other dog with an owner (who lived in a house in the center of Concepción) was Cachupín (which means mongrel dog in Chilean Spanish), who roamed around the city all day long. He would cross the Plaza Perú and surrounding area, as well as the university campus, thereby constituting one of the most notable cases of spatial ability, behavioral sociality and adaptability. The first author observed him preparing to use a crosswalk by looking left and right, in the right order, probably because of prior social learning. His age and expertise turned him into a leader for dogs and humans alike, and gave him an unmatched cognitive grasp of the territory. It should be noted that, just like the stray dogs, he never barked (cf. [10]’s finding that 3.34% of dogs only barked in extreme situations). In the course of his daily journeys, he had come to establish relationships with people and other stray dogs, almost becoming a go-between. Some stray dogs allowed people to pet them in Cachupín’s presence, even though these dogs were not looking for contact. Being stroked generates redirected and appeasing behaviors in pet dogs, signaling discomfort this way, and agreeing to be stroked is learned with humans [93]. University students knew him by name, as they saw him daily, not only on the lawns but even sleeping in some of their classrooms when it was cold. In addition, his noticeably sociable character caused not only humans to approach him, but also other dogs, including the very shy, who saw a chance to ask for food from humans. The anecdotes collected during our observations included the fact that Cachupín recognized the distinct sound of the chimes of the campus bell tower at midday, taking it as a signal to head toward the university, for he knew there would be more students moving around and consequently more opportunities for receiving handouts.

The two owned pet dogs, which wore collars with their name and the phone number of their owner, ranged across wide areas of the city center. This is why we categorized them as representing an extreme of the pet dog category (i.e., free-roaming pet dog) that has never previously been described in the literature [90,91].

To conclude on the Plaza Perú area, we found two more pronounced types of dogs’ sociability toward people, namely familiar stray dogs in a pack and free-roaming pet dogs. Furthermore, these two new categories of dogs, exemplified by the quiltros and Cachupín, interacted.

Our third finding was that the dogs’ distribution was related to the urban typology, albeit indirectly (i.e., through the relationship established between humans and dogs in each case). This can be clearly seen in Table 1 and Figure 3 where, for instance, guard dogs abound in residential areas and familiar stray dogs are to be found in packs in areas where there are more public spaces. Interestingly, this indirect link established a spatial occupation pattern with a degree of zoning, driven in some cases by people (pets living in apartments or houses and walked on a leash, guard dogs living in gardens and not walked in streets or beggars’ dogs staying close to their owner but off-leash in the streets), but in others by the free dogs (familiar stray or even feral dogs occupying various public spaces). While there may have been transitional areas (i.e., areas that could be traversed by various types of dogs), the free dogs were seldom seen occupying a space where there were dogs that had a different relationship with people from theirs. Thus, it is curious to note the very small number of familiar stray dogs that circulated in residential areas, and the absence of such individuals in the woods. Then again, this may have been because they had more difficulty obtaining food in residential areas and in the woods than in areas with large numbers of people passing through. Interestingly, the two free-roaming pet dogs exhibited similar spatial behaviors, crossing many types of urban areas and also showing exceptionally good spatial knowledge and navigational abilities. There is no literature yet on the navigational abilities of free dogs in cities, even when these are guide dogs. However, the abilities we observed are compatible with those of the dogs’ wild canid ancestors and with the spatial abilities highlighted in pet dogs in experimental conditions [94,95,96,97,98,99,100,101,102]. They are also compatible with the abilities taught and seen in guide dogs [103].

We, therefore, observed specific urban spatial typologies associated with the dogs’ patterns of distribution, as well as specific types of sociality with humans.

#### 3.1.1. Dogs Adapt Their Territory to the City

What comes out of these observations is the behavioral plasticity of dogs’ spatial abilities. The dogs we studied displayed eight different forms of adaptation, and two categories of dogs (familiar stray dogs and free-roaming pet dogs) did not stay in the areas where their nests were, but instead navigated between areas. Different motivations were probably at play: scavenging in the case of the stray dogs and sociality in the case of the free-roaming pet dogs.

The dogs’ knowledge of the territory made it possible to pinpoint the cognitive ability required, not only regarding the territoriality patterns the city offered, but also regarding dealing with its inhabitants (i.e., intra- and inter-specific social cognition). It was their learned experiences that configured the free-roaming dogs’ territorialities, with subtle variations between individuals.

In particular, based on our observations of the Plaza Perú pack and the fairly autonomous temperament of the three or four members of the pack who roamed on their own, we were able to identify a common set of basic activities that defined the dogs’ typical spatial daily routines. Instead of fixed habits or a daily sequence of actions, the familiar stray dogs of Plaza Perú exhibited a set of activities linked to the needs of their species (e.g., feeding, resting and mating; Table 2). These activities took place within a fairly circumscribed area and could come about in different ways. First, they could occur as a result of natural factors such as the season or the day’s weather (e.g., if it was raining or very hot) and took human activities into consideration (e.g., fewer pedestrians in summer, as students were on vacation, but more people on café terraces). Second, the dogs relied on their knowledge of how humans used the urban spaces on a regular basis (e.g., work patterns, meeting times in public or recreational spaces and rubbish collection times), and not only on a daily basis but apparently on a weekly one too (e.g., antiques fair on Saturdays and absence of students in the university sector on Sundays, which the familiar stray dogs avoided). Ethologists currently attribute this apparent temporal knowledge not to temporal cognition but to learning (i.e., an association of cues, as already mentioned above; see [9,40,104,105]). The set of activities, therefore, varied according to the possibilities the city had to offer (see also the pet dog–owner dyad in the city of Lyon, France [20]).

The familiar stray dogs relied on their nomadic mobility for survival [106], whereas the pet dogs and guard dogs had a more sedentary lifestyle that was, by definition, characterized by routines, just as they are in countries where dogs are leashed [20,107]. Even when we compared the behavior of the stray urban dogs with that of the feral dogs, the former had more complex and less predictable tendencies than the latter, adjusting them not only to natural elements like the feral dogs did but also to anthropic environmental conditions. The daily actions of the familiar stray dogs in a pack were even more unpredictable when the cue was not visible or known to the observer. Finally, it is worth mentioning that during mating periods, activities were altered not so much within the pack as in relation to the arrival of other stray dogs.

The familiar stray dogs, therefore, exhibited a peculiar degree of adaptation in this urban context, as well as all their usual sociality skills. Their territoriality patterns were built on adjustments and learning practices that enabled them to, for example, empty certain types of rubbish bins by tilting them (for dogs’ ability to manipulate objects to obtain food, see [21,108]), approach humans to obtain some kind of food either directly or indirectly, in the form of leftovers (learning, see [40]), or cross one-way and even two-way roads by looking both ways to see if cars were coming (social learning [83,84]) or follow a group of pedestrians using a crosswalk (for a recent review of experiments on dogs’ imitation of social agents, see [81]). We calculated that the familiar stray dogs in a pack crossed at least 8–10 streets per day.

Concerning the free-roaming pet dogs, we can assume that they were motivated to roam not only for food, but also for sociality [12,20,48,49,109]. Cachupín could only have limited interactions with leashed pet dogs and was actually seen with stray dogs. His territoriality was of a more extensive variety. His rounds covered several square kilometers in the city center, requiring considerable ability to handle all the city’s morphological barriers, such as the streets and traffic.

The older free-roaming pet dog had his own strategy for navigating in the city, as he walked slowly on the roadway, but was avoided by motorists. Unlike the familiar stray dogs and Cachupín, he followed the vehicles, not the sidewalks.

The abilities of the familiar stray dogs and the two free-roaming pet dogs resembled those of skilled guide dogs [103], as they had exhaustive spatial knowledge. The case of Cachupín is exceptional but not unique, for virtually every neighborhood of Concepción has a mature dog, often owned, who roams around.

These various canine practices, determined by the nature of the animal and the place, show that the territoriality patterns of the pack and the free-roaming pet dogs were constructed through learning processes. Practices may even have been transmitted between dogs belonging to the same pack from veterans to newcomers or incomers, or else acquired by mimicking human actions [83]. This mimetic process does not involve a relationship of dependency or the legacy of a past relationship with former owners, but instead an affiliative bond with humans (see [85,86,87,110]). This may benefit the dog in a context of shared interest [111], where humans derive entertainment and satisfy the strong cultural habit of accepting a canine presence, which can also warn of impending threats. All these cases in Concepción, therefore, illustrate varied canine territoriality patterns.

#### 3.1.2. Passages, Boundaries and Landmarks of the Urban Canine Territory

By observing the dogs of Plaza Perú, we were able to establish the different ways they used the space (i.e., territoriality patterns). These ways varied from categories of dogs listed by the literature, thereby generating spatial zoning [77] (Figure 3). The territories of the different categories of dogs varied according to their functions and reflected the diversity of the urban fabric. For instance, among the domesticated dogs, the guard dogs were concentrated more in a residential area consisting of houses, whereas the pet dogs were centered more on a residential sector featuring apartment blocks. Most of the feral dogs lived alone in wooded areas away from human contact, whereas the familiar stray dogs, which were more sociable, behave more nomadically and used all the spaces [112].

The passages (i.e., trails) corresponded mainly to those used by pedestrians: sidewalks and crosswalks. This illustrates how the dogs used humans as sources of reliable cues [85,86,87,110] and as models for navigating in the city (with the exception of the old basset hound, who followed vehicles in the road). This has never been reported in other cities. One possible explanation is that the flows of pedestrians and cars are more scattered in Concepción than in Istanbul, for instance. These free-roaming dogs, therefore, appeared to use the humans’ network of trails, possibly relying on smell to know which landmarks and trails to follow [113].

As indicated above, the familiar stray dogs displayed the widest variety of territorial uses, occupying a broader range of spaces than any other category we identified. In the case of the Plaza Perú pack, first, their nests were strategically located. The square was at the confluence of all the other areas. This square had many advantages: vegetation, good visibility for seeing and communicating, proximity to very different sectors that provided a variety of resources according to the time of day (i.e., according to human activity) and water from a nearby fountain. The pack of familiar stray dogs in Plaza Perú was thus very well positioned, even hierarchically with regard to pet or other stray dogs, and even people. For example, because of the square’s centrality, other packs of stray dogs from the city were forced to pass through, but never stayed unless the Plaza Perú pack allowed them to. In this sense, the territories of the different packs of stray dogs in Concepción were well defined but not hermetic, as the packs moved between territories. Plaza Perú is a nomadic middle territory for both species. It is a familiar and well-defined public space not just for humans but also for dogs, being used in similar ways by both. It also includes a small nest space for the local pack, with its own boundaries. Second, throughout the day, the pack moved beyond the nest space through other spaces dominated by other packs, generally without conflict. For instance, for part of the morning during the academic year, they would individually and independently move toward the campus and check rubbish bins, as well as areas frequented by students and other places they had identified as being food sources. In the afternoons, they would wend their way back and visit the terraces of cafés and restaurants, which were very busy at that time of day. Finally, at night they would seek shelter in Plaza Perú. As mentioned earlier, as they moved around their territory, they showed that they had learned complex actions such as crossing boundaries like avenues or streets. This knowledge protected them from other dogs, and at the same time enabled them to expand their foraging space.

The boundaries of the familiar stray dogs’ territories were permeable and overlapped [38]. For example, the popular and familiar stray dog Sir Perro from the university often used to be seen in Plaza Perú, showing that dogs from different packs can merge, see Sir Perro on the left in Figure 6.

The owned basset was also seen by HCM in Plaza Perú, coming from the city center. Cachupín was known to live around 600 m away, but was often seen around Plaza Perú and the campus. This means that he used to cross many streets. These many sightings in the square can be explained by the fact that there was enough food and, more generally, enough resources for all the dogs.

The boundaries and landmarks of the dogs living in houses or apartments were, of course, dependent on those of their owners [20]. The boundaries and landmarks of the free canines’ territory were perfectly superimposed onto the textures of the concrete urban fabric, reflecting a process of plasticity (e.g., [51,81,114,115]). There was, however, no evidence of local knowledge transmission or cultural transmission within each pack.

### 3.2. Snoopy from Yugoslavia Street in Hualpén

Another type of canine sociality was exemplified by a residential and very sociable stray dog (Type I) who had settled in a street. This social situation is typical of village dogs, except that here there was a single dog, rather than a group of dogs, living in a neighborhood. He acted as a companion and guard dog (i.e., a full pet) in the street environment, with two or three owners who fed him. In this context, not only did the stray dog gain the city as his territory, but the city also gained from his presence. Some types of stray dogs can leave such a strong impression on their urban neighborhood that they end up marking its identity.

#### 3.2.1. The Street Dog’s Domestic Neighborhood

Snoopy was a neighborhood dog in the residential sector of the municipality of Hualpén, in the Concepción metropolitan area (Figure 7), 8 km from Plaza Perú. He exemplified the behavioral adaptation of stray dogs in less central areas of Concepción, thus confirming the correlation between urban morphology and the adaptive behaviors of stray dogs, mediated by humans.

According to the neighbors, Snoopy was a stray dog who arrived on Yugoslavia Street in Hualpén in about 2011. Little by little, he gained the affection of the local residents, who fed him. As he had no specific owner, he ended up belonging to the whole street. As the months went by, he became a permanent fixture. He recognized all the residents, accompanied the children and elderly people to the bus stop, got along well with the residents’ pet dogs (and even cats), greeted neighbors when they came home from work, played with the children and even barked at strangers, especially at night. Like most of the dogs in Concepción, he also announced imminent earthquakes (very common in this region) by barking. Snoopy’s activities were feeding, resting and guarding (robberies, strangers, intruding dogs and earthquakes).

It should be noted that he would occasionally protect neighborhood children from unfamiliar stray dogs and even accompany neighbors who were drunk. All this reinforced his charisma among neighbors. Snoopy developed into a kind of night watchman and guardian of the street, thus adapting perfectly to his new habitat. Far from being an exception, this is quite a normal situation, not only in the residential areas of Concepción, but also in other Chilean cities and even other Latin American countries. The street became his territory, which he almost never left, except when accompanied by one of its residents, and then only exceptionally. He behaved as if he was the street’s guard. The nomadic behavior that had brought him to the street became more sedentary, as a result of his social adaptation (i.e., close social relationship with the neighborhood). We categorized him as a pet-stray dog (i.e., Type II), but he was an extreme example, owing to the extent and quality of his relationships with humans. He was thus a visible symbol of the relationships between local residents, and although nobody ever completely took ownership of him, he was recognized as an inhabitant of the street and a factor for social cohesion. For all these reasons, we regarded him as a pet-stray dog.

#### 3.2.2. The Dog as a Component of the Neighborhood’s Identity

The neighborhood’s close relationship with Snoopy could be explained by his affiliative behavior within the community. Additionally, the experiences and daily occurrences contributed; they made it possible to identify him with the neighborhood and more particularly with the street, as part of a vernacular heritage [116]. Snoopy ended up representing the experiences of the neighborhood. However, his charismatic presence and the affection he inspired eventually led to his poisoning in 2013 by one of the street’s female residents, who hated not so much the dog as what he represented. This neighbor detested the other residents and finally took her revenge through the animal. This is one more example of the key role played by stray dogs in the construction of neighborhood identity. Snoopy’s demise also illustrates the harsh reality of the lives of most of these dogs, who have a reduced life expectancy owing to accidents, illness and the scarce and isolated violence that is meted out to them. Despite being an anonymous dog with a previously untold story, he has been immortalized in an image of the street on Google Earth’s Street View. This image sums up the dog’s referential function, and even today, the street’s residents look at it with fondness. This case study demonstrates how stray dogs can adapt to the urban morphology and become a collective identifier for human inhabitants.

It is the same story for the famous familiar stray dog Sir Perro, who lived outside the library on the university campus. A Facebook page, which still exists, had been opened in 2010 to collect money for his veterinary care and he became the students’ mascot. His death was announced in the news in January 2017 [117] in the same way that it would have been for a human resident of the neighborhood. Both Snoopy and Sir Perro helped to provide citydwellers with a spatial and social identity, as well as a deeper sense of place [118].

## 4. Discussion

### 4.1. Diverse Adaptations of Dogs to Diverse Urban Morphologies

The stray dogs of Concepción had a behavioral peculiarity. In this urban context, the dogs who survived lived in a state of nonhuman dependency, exhibiting a less domesticated (or less tamed) social form (i.e., no learning imposed by humans). Consequently, whereas cities encourage anonymous and individualized lifestyles among humans [119], Concepción allowed these dogs to display all their (untamed) forms of sociability [10,14,15,16,17,18,19,20,21,22,23,24,25,26]. We found that the territories of the different dogs could be almost perfectly superimposed onto the manmade city’s territorial patterns, and the dogs exhibited a range of territorialities [120]. This was probably made possible by Chilean culture (see Appendix C). To sum up, each category of dog maintained a degree of spatial distance from humans and a carefully scaled relationship with them [31]. Dogs belonging to the new categories we identified (familiar stray dogs and free-roaming pet dogs) crossed all kinds of spaces in the urban context, but for dogs in the other categories, spatial behaviors/territorialities were dictated by humans. Guard dogs, for instance, were more present in residential and peripheral areas, were kept in a limited space that they had to protect, and were allowed very few contacts with other dogs. Pet dogs were based inside houses or apartments, and their closest relationships were mainly with their owners, although they had quite good interactions with other dogs—mainly other pet dogs, but possibly also a few familiar stray dogs when they walked with their owners. They certainly shared some of the public spaces with the familiar stray dogs and had almost the same movements, but displayed different attitudes. In the case of feral dogs, there was no contact with other kinds of dogs or humans: they avoided anthropized spaces, occupied woodland and remained on the periphery of city activities. They hid from humans in a nonaggressive way, were solitary and skinny, suggesting that they had an anthropized diet, but one that was less copious than that of the dogs in packs in the city center. The dogs of Concepción, therefore, displayed a wide and varied urban distribution, depending on their behavior toward humans, with different attitudes displayed by different categories (no relations, few relations or close relations).

In the case of Concepción, we, therefore, discovered a wide variety of canine behaviors and roles, with diverse levels of sociability among the dogs [121] and with humans. The dogs’ social adjustment was linked to the urban context (see also [9] for coyotes and foxes in Madison, who each had qualitatively different home ranges—a spatial partitioning that probably promoted positive coexistence between these sympatric canids in urban areas). Human cultural rules and the different types of spaces led to a wide variety of adaptations by these dogs, even though they all had the same basic needs. Urban dogs not just in Concepción but throughout Chile represent an excellent laboratory for research that could also be extended, albeit to a lesser degree, to other South American countries, or even to other areas of the world. Even in the Latin American context, the sheer variety of Chilean urban dogs constitutes a specificity. Both types of dogs in the newly identified *familiar stray category* exhibited closer relationships in terms of social distance (e.g., occasionally being fed directly by humans and sometimes stroked, using crosswalks either with pedestrians or on their own, having learned how to do so from humans) than stray dogs in other parts of the world (e.g., Asia, Africa or likely most of the dogs of Turkey/Istanbul). This very specific situation can probably be traced back to a bygone rural tradition and/or to the Mesoamerican civilization, where humans maintained a specific relationship with animals, seeing them as part of the same natural world [3]. Even within Chile, the presence of stray dogs varies from north to south, according to cultural traditions [54]. The best examples are to be found in some of the Southern-Central Chilean harbor cities, such as Concepción-Talcahuano, Valparaiso or Puerto Montt, as the presence of dogs is less marked in big city centers like that of Santiago. In Valparaiso, for example, there are so many stray dogs that they have taken on an iconic status (Figure 8).

The present study carried out in the Latin American context, and more specifically the Chilean reality, found that the two species (i.e., humans and dogs) shared the public space, mostly without conflict. They nevertheless exhibited different territorialities, and there was little direct contact between them. That said, there were many instances of indirect dependence. For instance, stray dogs were such a common sight that they had almost become part of the urban landscape. In fact, people moved around without paying any attention to the dogs, although this does not mean that they ignored them. For example, if a dog was stranded in the middle of the street or sidewalk, passers-by and motorists would dodge the animal without having even minimal contact with it, and vice versa. However, the dogs’ greatest adaptation (i.e., showing how human lifestyles had influenced the dogs’ habits and daily activities) concerned food. The stray dogs’ itinerary through the city matched human work schedules. Dogs moved to and approached places and people related to food as if to see whether they could obtain nourishment. For example, dogs would approach the university in the morning as well as at midday, having learned that the students arrived with snacks. We also observed that some dogs settled near food stalls at peak times (i.e., when it was crowded). All these patterns make it possible to talk about parallel lives, with indirect relations between stray dogs and humans.

### 4.2. The City as Life Learning: Territorialities

The case of the Concepción dogs highlights a living and dynamic reality of the city, characterized by a close interrelationship between people, animals and the city itself. The present study sheds light on how an anthropized environment unexpectedly influences dogs’ territorialities and socialities. It highlights the effect of the city’s morphology and human activities on the learning processes of stray dogs (e.g., use of human behaviors as cues). Likewise, these stray dogs contribute to human social urban identities, as illustrated by the case of Snoopy in Hualpén and the mere presence of the stray dogs.

Urban dogs’ behaviors are good examples of adjustment processes. Our observations enabled us to show the plasticity of dogs’ learning processes (e.g., territoriality) in both their individual and collective attitudes [1,39,40,41,42,43,44,45,46]. Urban landscapes, associated with human interactions, require a broad range of potential adjustments from dogs. The present study identified the behavioral adjustment of dogs in general, and stray dogs in particular, to a specific and quite unusual city.

These stray dogs and pet dogs living in cities exemplify the ultimate adaptation of dogs on Earth, where they are virtually ubiquitous [26,39]. According to Coppinger and Coppinger [51], niches are never stable, meaning that “what we think of as being a species now was different in the past and will be different in the future. Evolution is the process of species continuously evolving to new and changing niches”.

## 5. Conclusions

We found three new types of dog socialities (cf. Table 3), reclassifying by the way dog groups in relation to different types of urban areas and the present specific human culture permitting this. To conclude, further research is now needed to reconsider the meaning of the spatial and social relationships of all living beings and not just humans to their environment. Our findings open up a fascinating avenue of research for both ethologists and geographers, involving a more pluralistic perspective: for although the presence of dogs in cities is certainly nothing new, studies are still scarce. Exploring the social and spatial settings of dogs in cities provides a means of identifying all the possible sociospatial organizations (i.e., territorialities and territories) of *Canis familiaris*. Of course, our results indicate the potential of dogs, in general, to live in good harmony with people, provided appropriate behaviors regarding them.

The relationship between humans and dogs in a city landscape should be studied not just with the purpose of learning more about dogs’ behavioral plasticity or territorialities (i.e., social and spatial learning processes and adjustments), but also from an anthropic (i.e., human-centered) perspective. Dogs, especially the stray dogs in Concepción and other Chilean cities, contribute to humans’ urban social identity. In Chile, stray dogs are part of the urban landscape and have a place in local neighborhoods and, on a larger scale, in the country’s urban reality. This sharing of urban space results from a learning process not just among dogs, but among humans too. Here, the city is a cultural space with a wide range of diverse situations for dogs, for humans and for the human–dog relationship, a product of dogs’ sociality and their dependency on humans. The human–dog relationship we observed in Concepción differed from that observed in other Western urban contexts because, as we mentioned earlier, it was based on a more tolerant attitude on the part of humans, such that one-third of the dogs in this human context were free-roaming. This specific situation provided a very interesting study area where we could not only learn more about dogs but also delve deeper into their learning processes and examine the way they adjust their relations with humans [1,39,40,41,42,43,44,45,46].

The present research yielded findings about the cognitive, social and spatial adjustment abilities of dogs and their territorialities, informing the debate about the place of animals in geography. We confirmed our hypothesis: we found numerous types of sociability in this city, both new and known ones. At an ethological level, we confirmed the presence of previously established categories: family dogs (pet dogs, beggars’ dog), restricted dogs (guard dogs) and stray dogs (dogs in Lenga that had no familiarity with humans, aggressive dogs at the far end of the campus and feral dogs in the woods). We also identified three new categories: (1) familiar stray dogs in packs (pack of three individuals in Plaza Perú) that are spatially and socially very close to humans (extreme example of Type II; dotted underlining in Figure 9; see also Sir Perro); (2) pet-stray dogs, that is, village dogs enjoying very close interactions with people (e.g., Snoopy, an extreme example of Type I; dotted underlining in Figure 9); and (3) free-roaming pet dogs (the owned dogs Cachupín and the old basset; solid underlining in Figure 9). The dogs in these three categories displayed different socialities and degrees of sociability, probably reflecting tolerant human attitudes toward dogs and varieties of urban areas (Figure 9). We showed how their spatial and social learning skills constitute a means of survival and adjustment (territorialities). Finally, these new categories lie at the very heart of the debate about early domestication (and taming). They blur the implicit classic Western divide between nature and culture, confirming earlier statements by Nesbitt [37] and Boitani et al. [29] that domestication and taming are a matter of degree, affected by anthropogenic factors, and showing that dogs can shift from one category to another in the course of their lives. The dog species thus exhibits a considerable potential for social and spatial adjustment. The fact that it depends on the spatial layout and human culture of their environment, explains the presence of dogs wherever humans are. Furthermore, it has implications for coping with the presence of dogs in numerous and varied human societies.

## Figures and Tables

**Figure 1 animals-10-00298-f001:**
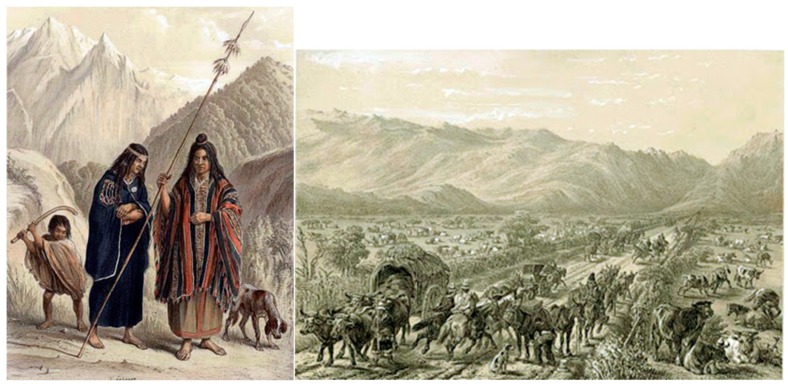
**Left**: Dogs were a common presence in Mesoamerican cultures (Mapuche). **Right**: Dogs accompanied the first European settlers in the 16th century [3].

**Figure 2 animals-10-00298-f002:**
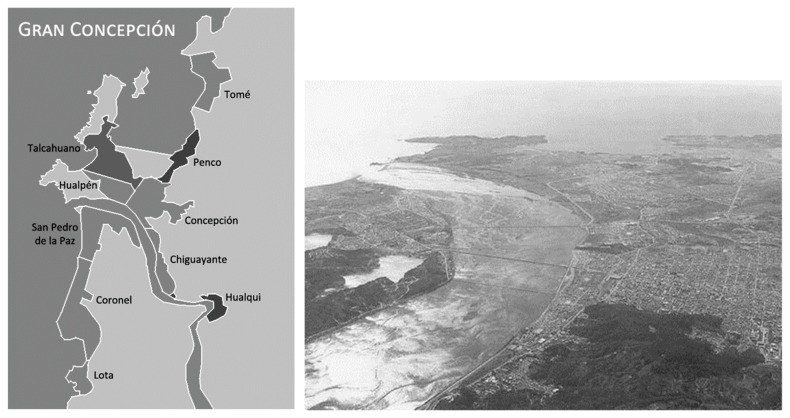
Study area location (**left**) and a general view of the metropolitan area of Concepción (**right**). Sources: https://es.wikipedia.org/wiki/Gran_Concepción, accessed on 21 December 2019.

**Figure 3 animals-10-00298-f003:**
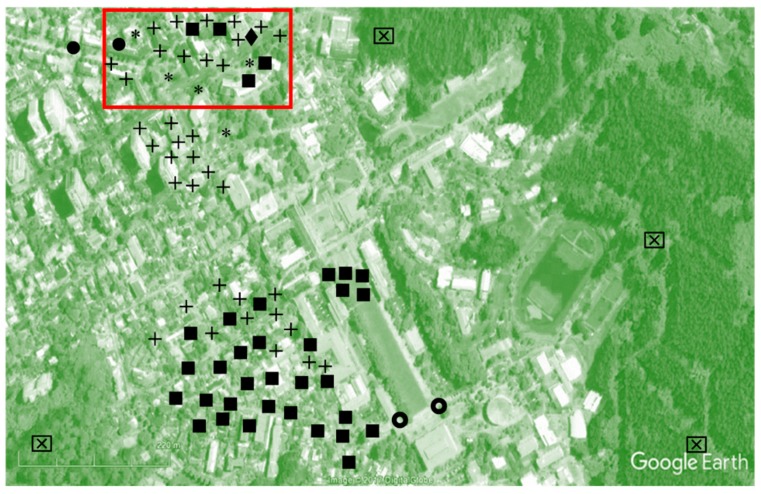
Distribution of dogs in the Plaza Perú area, Concepción. See Table 1 for the key to the signs. Source: Google Earth.

**Figure 4 animals-10-00298-f004:**
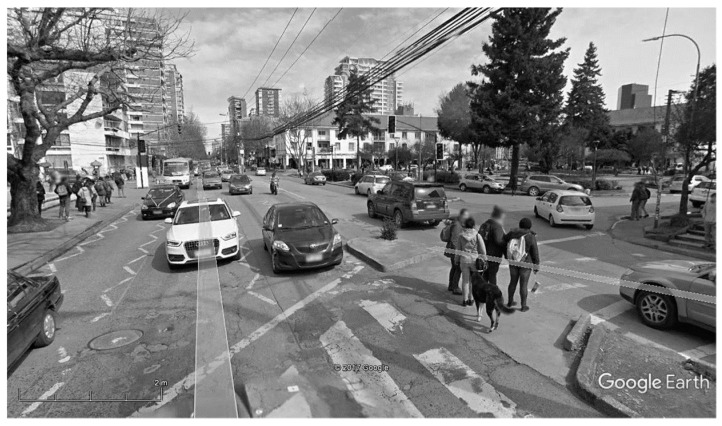
Crosswalk in Plaza Perú with a free-roaming dog on a traffic island between two roads crossing with people. Source: www.google earth.com.

**Figure 5 animals-10-00298-f005:**
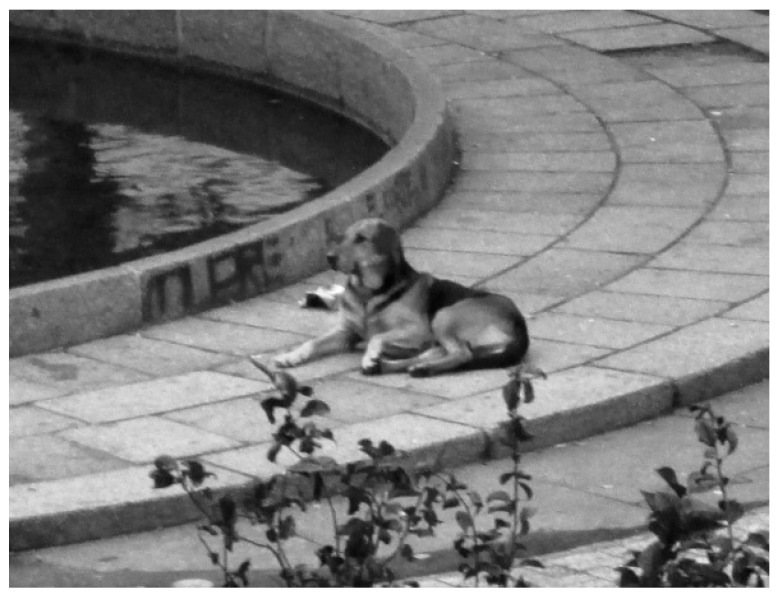
One of the dog packs in Plaza Perú, shared by people and dogs. Source: HCM.

**Figure 6 animals-10-00298-f006:**
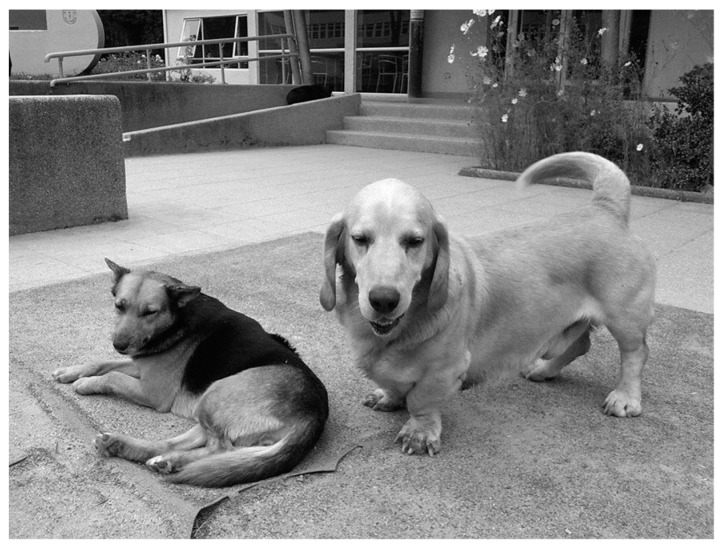
The famous free-ranging dog Sir Perro, and the basset. [92].

**Figure 7 animals-10-00298-f007:**
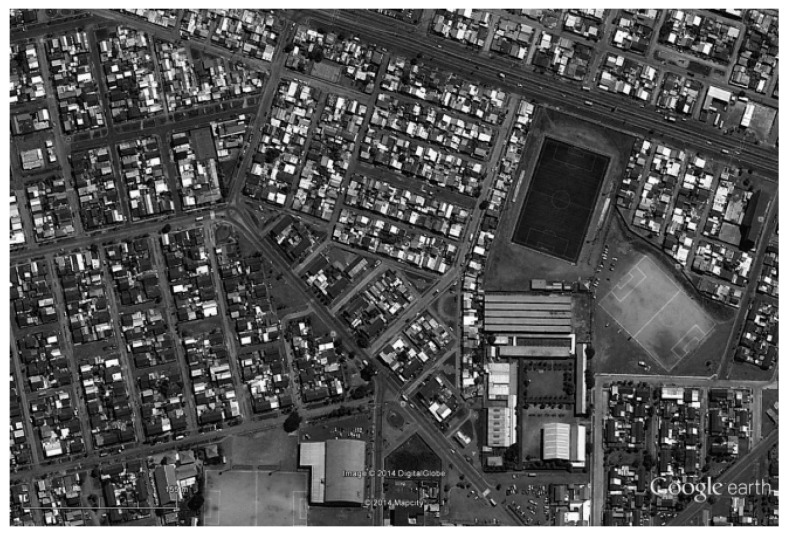
The spatial arrangement of the area including Yugoslavia Street, Hualpén. Sources: Google Earth, accessed 21 December 2019.

**Figure 8 animals-10-00298-f008:**
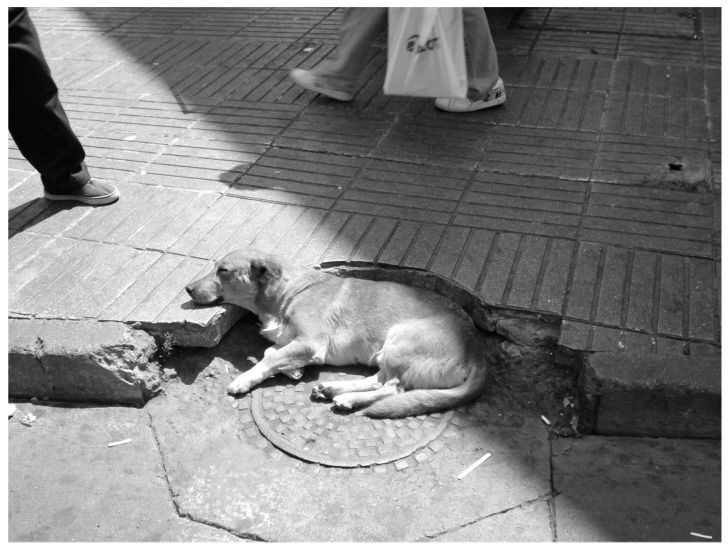
Stray dog in Valparaiso, lying on the street. Source: HCM.

**Figure 9 animals-10-00298-f009:**
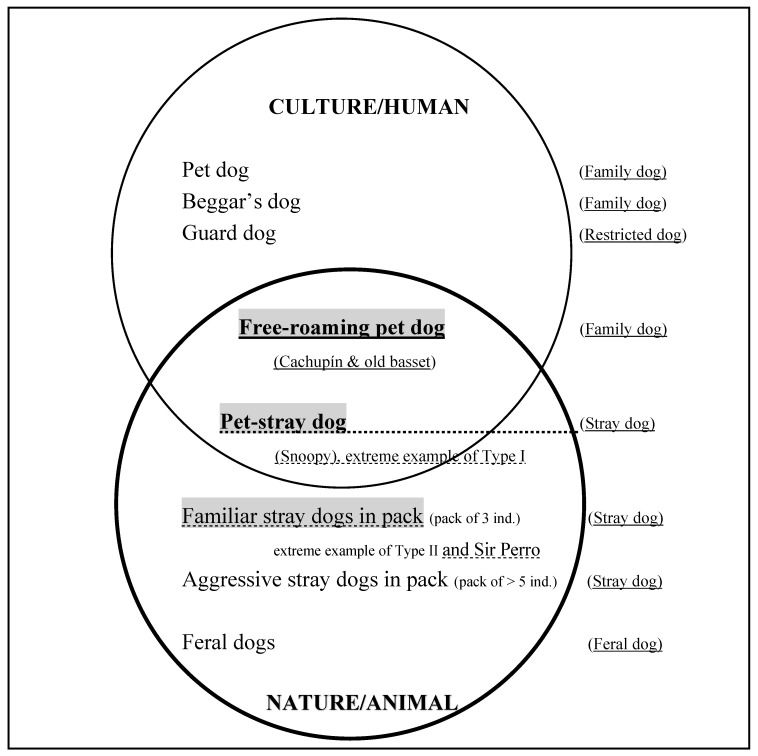
This schema summarizes our categorization of the dogs in Concepción, with those most closely related to humans and culture (pet dogs) at the top, and those most closely related to nature and its wilderness (feral dogs) at the bottom. The intermediate categories correspond to the dogs’ different social and spatial adjustment abilities: they are defined by the closeness of their social and spatial relationships with humans, with discrete categories for the Culture/Human type of relationship (upper circle: effect of human culture/enculturation) and for the Nature/Animal type of relationship (down circle: no effect of human culture/primacy of the natural characteristics of the canid species). The schema includes the three new categories we identified (in grey): two variants of stray dogs (dotted underlining), and one variant of pet dogs (solid underlining), see also Table 1.

**Table 1 animals-10-00298-t001:** Type of sociality or category of dogs, location and number of individuals for each canine category observed in or around Plaza Perú area. Despite being from Hualpén, 8 km away, the dog Snoopy is included in this table (see the section dedicated to him). The four categories in italics are variants of pet dogs. Source: HCM.

Category of Canis Familiaris	Number of Dogs, Location(s) and Behavioral Main Characteristics
₊	*Pet dog* (Family dog)	36 dogs living at home with their owner (leashed)
♦	*Beggar’s dog* (Family dog)	1 dog living unleashed with its beggar owner in the street
★	*Abandoned dog* (Family dog)	1/week
■	*Guard dog*(Restricted dog)	25 dogs in gardens that barked at people and other dogs5 leashed dogs owned by security guards on the university campus4 unleashed dogs in Plaza Perú owned by a night watchman
●	Free-roaming pet dog(Family dog)	1 pet dog that had an owner but wandered around the neighborhood all day long (Cachupín)1 old basset pet dog that had an owner and wandered around the neighborhood all day long (Napoleon)
Not in Figure 4/in Hualpen	Pet-stray dog(Stray dogs)	1 stray dog that belonged to a street and exhibited pet dog behaviors (multiple neighborhood caretakers) (Snoopy)
*	Familiar stray dog in a pack(Stray dogs)	5 packs of 3 individuals, including Sir Perro
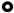	*Aggressive stray dog in pack*(Stray dogs)	2 packs of 5 individuals that barked at people and other dogs and bit them, as though defending the area around the library
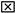	Feral dog (Feral dogs)	4 in woods (fully solitary)
	Total number of individuals	103

**Table 2 animals-10-00298-t002:** Typical daily activities of the stray dogs in Plaza Perú.

Time of Day	Typical Daily Activities of the Familiar Stray Dogs
Morning	Sleeping in the sun (more during weekends) Moving away from the square Searching for food when the students arrived on the university campus Going through rubbish bins Visiting strategic locations: streetfood stalls, presence of pedestrians, bakery entrances
Midday	Touring the surrounding area Sleeping in the sun
Afternoon	Returning to the square. Visiting café terraces in search of food Visiting the central areas of the square in search of food when the students left the university and gathered in groups
Nightfall	Taking refuge in sheltered nests (holes) in the square

**Table 3 animals-10-00298-t003:** Listed categories of dogs in the literature (left column) and the three new categories of dogs evidenced in the present research (right column).

Listed Categories of Dogs in the Literature	New/Additional Categories of Dogs Evidenced
Restricted dog	Guard dog	
Family dog	Pet dog	
Beggar’s dog
Abandoned dog
	Free-roaming pet dog (Cachupín)
Stray dog		Pet-stray dog (Snoopy)
	Familiar stray dog in a pack (quiltros)
Aggressive stray dog	
Feral dog

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
