# Peer review of "Coexistence of Diversified Dog Socialities and Territorialities in the City of Concepción, Chile"

_animals, 2020, doi:10.3390/ani10020298_

Round 1

Reviewer 1 Report

Thanks for adding the aim (lines 136-143). That helps guide the intent of the paper.

The research provides a good example that street dogs are not 'mean' dogs as their reputation sometimes dictates. Were there any negative incidents (fighting among dogs, resource guarding, growling at passers by...)?

Author Response

See word document uploaded

Reviewer 2 Report

The attempt to describe various types of dogs living Concepcion is very interesting, but some trimming and clarification are needed with additional quantitative data.

One of the characteristics of family owned dogs are that  reproduction is supervised by the owner (Miklosi 2018), but it is not included in the definition of the dogs categorized in this paper.  If it is excluded for certain reason, then the reason for not including  may need to be stated clearly.  

The introduction mentions about various studies on free ranging dogs, but are not discussed deeply enough to related to the contents of the paper.   Please discuss of the studies which are closely related to designing this research.

Method section needs to be more precise and compact. 

Results section should only state the results, but in this paper, extensive interpretation of the data is also stated. For example, would it be possible to show quantitative data to support the contents of Table 2? Interpretatino of data should be  in the discussion section.  

Overall trimming and also addition of quantitative data to prove the points presented in the results and discussion. 

As for the ethical concerns, it is up to the editorial board to decide conducting ethnographic observation in this manner is permitted or not.

Author Response

See word document uploaded

Reviewer 3 Report

I suggest to improve:

the reference on the method used to evaluate the behaviors of dogs some technical terms as indicated in my review

Author Response

See word document uploaded

Round 2

Reviewer 2 Report

Information provided in this paper is very valuable with recent increase in papers on free ranging, free roaming or street dogs around the world.  I look forward to upcoming researches of dogs in South America inspired by this paper.

Author Response

Thank you very much for your help.

This manuscript is a resubmission of an earlier submission. The following is a list of the peer review reports and author responses from that submission.

Round 1

Reviewer 1 Report

The topic is so relevant and in-demand; however, the way that it is currently written, much of the content is lost due to run-on sentences, grammatical inconsistencies and incorrect citation formatting. Once this has been reviewed by an editor, I am happy to review content. Thanks!

Reviewer 2 Report

The authors would need to be careful in how they classify the dogs. It is unclear how they have knowledge of the origins of each dog they observed and how they avoided multiple observations of the same dogs.

The methods are not nearly clear or detailed enough to determine how the dogs were identified and how their distribution in space was determined. Following the dogs may have altered their movements and use of space. It sounds like the observations were not distributed across the city landscape. The sampling thus seems limited. It is not clear how the observations were systematic or consistent over time. It is not clear how the social behaviors were defined or recorded. The results are not presented in a scientific manner with no attempt to quantify or classify variables of interest.

The sentence on lines 20-22 and the first line of the abstract don’t make sense to me.

Some of the writing is very awkward (e.g.., “one third of them seen were.. line 24) and contains some typos (e.g., line 92).

The paper is far too long for the contribution it makes.